**Subject Category:**
Biology (whole organism)

microbiology/environmental science

biochar, paddy soil, purple soil, soil carbon pool, soil respiration, soil microorganism

**Author for correspondence:**
Guoshun Liu
e-mail: ysh.24@163.com

# Biochar application on paddy and purple soils in southern China: soil carbon and biotic activity

Shen Yan[1,2], Zhengyang Niu[1,2], Aigai Zhang[1,2], Haitao Yan[1,2], He Zhang[3], Kuanxin He[4], Xianyi Xiao[4], Nianlei Wang[4], Chengwei Guan[4] and Guoshun Liu[1,2]

[1]Department of Tobacco cultivation, Tobacco Cultivation Key Laboratory in Tobacco Industry, Zhengzhou 450002, People's Republic of China
[2]College of Tobacco Science, Henan Agricultural University, Zhengzhou 450002, People's Republic of China
[3]School of Environmental Science and Engineering, Donghua University, Shanghai 201620, People's Republic of China
[4]Department of Tobacco cultivation, Tobacco Science Institute of Jiangxi Province, Nanchang 330025, People's Republic of China

GL, 0000-0003-2437-5643

Soil carbon reserves are the largest terrestrial carbon pools. Common agricultural practices, such as high fertilization rates and intensive crop rotation, have led to global-scale environmental changes, including decreased soil organic matter, lower carbon/nitrogen ratios and disruption of soil carbon pools. These changes have resulted in a decrease in soil microbial activity, severe reduction in soil fertility and transformation of soil nutrients, thereby causing soil nutrient imbalance, which seriously affects crop production. In this study, 16S rDNA-based analysis and static chamber-gas chromatography were used to elucidate the effects of continuous application of straw biochar on soil carbon pools and the soil microbial environments of two typical soil types (purple and paddy soils) in southern China. Application of biochar (1) improved the soil carbon pool and its activity, (2) significantly promoted the release of soil $CO_2$ and (3) improved the soil carbon environment. Soil carbon content was closely correlated with the abundance of organisms belonging to two orders, Lactobacillales and Bacteroidales, and, more specifically, to the genus *Lactococcus*. These results suggest that biochar affects the soil carbon environment and soil microorganism abundance, which in turn may improve the soil carbon pool.

# 1. Introduction

The soil carbon pool, which consists of organic and inorganic carbon, is the largest carbon pool in the terrestrial ecosystem, and its total reserve is approximately 3.3 times that of the atmospheric carbon pool [1]. Soil organic carbon (SOC) content is often regarded as an important index for evaluating the potential fertility of soil [2,3], and its dynamic equilibrium has a direct impact on soil fertility and crop yields. Human activities, such as high rates of fertilization and intensive crop rotation systems, have resulted in decreases in soil organic matter and carbon/nitrogen ratio, and imbalances in the soil carbon pool on a global scale [4,5]. Overall declines in soil microbial populations and microbial disequilibrium [6,7] have severely reduced the nutrient availability and nutrient transformation capacity of the soil. These changes have resulted in substantial nutrient imbalances in the soil environment [8], which have affected crop production.

Biochar usually refers to the highly aromatic organic matter derived from the pyrolysis of any solid biomass. It can persist in the environment and plays an important role in global biogeochemical cycling, climate change and environmental systems as a part of the SOC pool [9,10]. Biochar is also considered to be an important reservoir for atmospheric $CO_2$ [11]. Further, being a possible source of the highly aromatic component of soil humus, biochar has a crucial role in maintaining and increasing the SOC pool and retaining soil nutrients, improving soil fertility and maintaining soil ecosystem balance [12,13].

Soil microbial communities are known to be acutely sensitive to soil environmental changes [14]. Biochar is alkaline and porous, has a high specific surface area and numerous negative surface charges and includes high-charge dense materials [15]. An increase in biochar content in the soil can change the soil environment and microbial habitat, thereby affecting the biogeochemical cycling of soil carbon. As a substance with a high carbon content, adding biochar to the soil will directly supplement the organic carbon sources needed for soil carbon cycling [16,17]. The high stability of biochar results from its complex aromatic structure and physical and chemical protective effects. When biochar enters the soil, its stable group greatly enriches the SOC pool and is stored in the soil for a long time, enriching the total amount of organic matter in the soil [18]. The labile components of biochar, such as aliphatic carbon speciation matter, can supplement the soil carbon pool in the form of soluble organic matter [19,20]. Biochar addition can improve soil water-holding capacity [21], reduce soil bulk density, promote soil ECE (Cation Exchange Capacity) and pH [22], change soil biochemical reaction conditions and stimulate soil enzyme activity [23] and promote soil microbial reproduction [24].

In addition, biochar application has an obvious effect on soil mineralization [25]. In recent years, many studies have highlighted that biochar application can stimulate the soil and promote the mineralization of soil organic matter. This is mainly because biochar application provides a certain quantity of easily biodegradable organic substances for soil microorganisms, causing soil microbial co-metabolism and stimulating the mineralization of soil organic matter [26]. However, some studies have shown that biochar (such as pecan-shell made in the conditions of 700°C) can inhibit the mineralization of soil organic matter [27]. This is mainly due to the fact that biochar's bulk physical and chemical properties vary as a function of type and temperature conditions [28], and the adsorption of the porous structure of biochar organic carbon on soil organic matter, inhibiting the degradation of soil microorganisms and, thereby, reducing mineralization efficiency [29].

China has an annual output of 700−800 million tons of straw, but its utilization rate is less than 50%; more than 30% of the straw is discarded or burned, resulting in resource wastage and environmental pollution [30,31]. China has a long history of biochar fabrication and application [32]. Conversion of crop straw into biochar is carried out by straw carbonization, which can then be returned to the soil. It not only avoids the burning of straw but also increases carbon sequestration of soil. Thus, it is an effective means for comprehensive utilization of straw resources. In recent years, a few studies have investigated the effects of straw biochar on soil carbon cycling in southern China [33,34]; however, straw biochar in tobacco field application is heretofore uninvestigated. Therefore, in the present study, field experiments were conducted in two typical soils (paddy soil and purple soil) in southern China to evaluate the effects of the quarterly return of straw biochar on carbon sequestration and soil carbon pools.

In order to study the effects of biochar on the soil carbon pool environment, 16S rDNA-based analysis and static chamber-gas chromatography were used to examine the effects of 2 years of continuous biochar application on soil carbon pool components, microorganisms and soil respiration in southern China. The findings of this study provide a new scientific basis and direction for the application of straw carbonization and return in southern China.

**Table 1.** Nutrient status of the experimental soils.

| soil type | organic matter (mg · g$^{-1}$) | hydro-N (mg · kg$^{-1}$) | available P (mg · kg$^{-1}$) | available K (mg · kg$^{-1}$) | pH | total C (mg · g$^{-1}$) | total N (mg · g$^{-1}$) |
|---|---|---|---|---|---|---|---|
| paddy soil | 27.04 | 129.80 | 24.20 | 109.51 | 5.59 | 16.60 | 2.10 |
| purple soil | 9.67 | 43.31 | 7.09 | 223.54 | 7.30 | 15.00 | 0.70 |

# 2. Material and methods

## 2.1. Study site

The study was conducted at a tobacco production field in Xiniu town, Xinfeng County, Ganzhou city, Jiangxi Province (paddy soil: 25°27′11.71″ N, 114°51′54.25″ E; purple soil: 25°26′48.33″ N, 114°51′28.82″ E). The area is characterized by a subtropical moist monsoon climate: average annual sunshine hours are 1473.3–2077.5, with an average annual temperature of 18–19.7°C, frost-free period of 250 d and an average annual rainfall of 1410–1762 mm. The parent material of the paddy soil is fluvial alluvium, with soil subtype bleaching iron stagnic anthrosols, and a section depth of 110 cm; the parent material of the purple soil is purple mud (page) rock weathering residue—clinosol, subtype calcareous purple orthent, with a section depth of 25 cm. The nutrient status of the study sites is shown in table 1.

## 2.2. Experimental design

In 2015 and 2016, biochar was applied to the soil surface at a rate of 7200 kg h m$^{-2}$ biochar per year for 1 year (treatment group T1) or two years (T2) and mixed with the topsoil (0–20 cm). The control (CK) soil was left untreated and other measures were the same. The amount of biochar was determined based on the findings from previous unpublished studies, in which the optimum dosage affecting soil characteristics and the yield and quality of tobacco were revealed. Each experimental plot had an area of 666 m$^2$ and each treatment was replicated three times, in a random block arrangement. The tobacco variety Yunyan 87 was transplanted to the study site in early March 2015, with a row spacing of 50 × 120 cm. Local high-quality tobacco production management was used. The cropping systems on the paddy and purple soils were rice–tobacco rotation and bean–tobacco rotation, respectively. Tobacco was cultivated until 105 d after transplanting (tobacco harvest). The fertilization regimens for the two soils (paddy and purple) were 135 kg h m$^{-2}$ and 142.5 kg h m$^{-2}$ of pure nitrogen at N:P:K of 1 : 0.9 : 2.8 and 1 : 1 : 2.8, respectively. The biochar used (electronic supplementary material, table S1) was provided by Sanli New Energy Co. Ltd.; it was produced from rice straw via pyrolysis between 350 and 480°C [28], with a technology of continuous production in a vertical kiln [35].

## 2.3. Sample collection

Soil samples were collected after planting 60 d in 2015 and 2016, and soil respiration was also determined. Five representative strains of tobacco from each plot were pulled from the soil, including roots and soil. Surface soil was shaken off the plants and soil that adhered to the root surface was classified as rhizosphere soil [36,37]. Soil samples from the 0–20-cm and 20–40-cm soil layers were also obtained. Total carbon (TC), total organic carbon (TOC), microbial biomass carbon (MBC), easily oxidized carbon (EOC) and dissolved organic carbon (DOC) content were measured for all soil samples. Half of the samples were screened using a 10-μm mesh sieve and refrigerated at 4°C without air-drying; analyses of relevant indices were performed on the remaining samples after they had been air-dried and screened using a 20-μm mesh sieve. All 15 samples of tobacco rhizosphere soil were mixed to determine soil microbial population abundance.

## 2.4. Method for the analysis of soil samples

Soil TC and total nitrogen were determined using a C : N elemental analyser (Vario MAX; Elementar, Germany). Organic carbon was assessed using a Vario TOC Cube (Elementar), and fumigation and

extraction methods were performed to determine MBC following Vance *et al.* [38]. The EOC was detected using $KMnO_4$ (333 mM) oxidation as described by Blair *et al.* [39]. Water-soluble carbon was extracted from 30 g of refrigerated soil sample dissolved in 60 ml of deionized water. The mixture was shaken at 250 rpm at 25°C for 30 min and then centrifuged for 10 min at 4000$g$ [40]. Next, the supernatant was filtered through a 0.45-μm Millipore filter. Water-soluble carbon in the extracts was detected using an automated TOC analyser (Vario TOC Cube; Elementar).

## 2.5. Metagenome analysis of soil microbial community structure

### 2.5.1. DNA extraction and PCR amplification

Microbial DNA was extracted from the 0.5 g soil samples using the E.Z.N.A. soil DNA kit (Omega Biotek, Norcross, GA, USA) according to the manufacturer's protocols. The V3–V4 region of the bacterial 16S ribosomal RNA gene was amplified using PCR. PCRs were performed in triplicate in 20 μl mixtures containing 4 μl of 5× FastPfu Buffer, 2 μl of 2.5 mM dNTPs, 0.8 μl of each primer (5 μM), 0.4 μl of FastPfu Polymerase and 10 ng of template DNA. The PCR conditions were as follows: 95°C for 2 min, followed by 27 cycles at 95°C for 30 s, 55°C for 30 s and 72°C for 30 s, and a final extension at 72°C for 5 min using primers 341F (5′-CCTAYGGGRBGCASCAG-3′) and 806R (5′-GGACTACNNGGGTATCTAAT-3′). A barcode of 8-bp sequences unique to each microbial sample was also included. The amplification yielded an expected sequence length of around 465 bp.

### 2.5.2. Illumina MiSeq sequencing analysis

Amplicons were extracted from 2% agarose gels and purified using the AxyPrep DNA Gel Extraction Kit (Axygen Biosciences, Union City, CA, USA) according to the manufacturer's instructions and quantified using QuantiFluor-ST (Promega Corporation, Madison, WI, USA). The purified amplicons were pooled in equimolar quantities and paired-end sequenced (2 × 250 bp) on an Illumina platform according to standard Illumina protocols. The operational taxonomic units (OTUs) were obtained from tags using a cluster analysis, and a Ribosomal Database Project classifier was used for accurate annotation of the tags and OTUs.

### 2.5.3. Bioinformatics analysis

Raw data containing adapters or low-quality reads would affect the subsequent assembly and analysis. Thus, to obtain high-quality clean reads, raw reads were further filtered by removing reads that (1) contained more than 10% unknown nucleotides (N); or (2) contained less than 80% of bases with quality (Q-value) greater than 20. Paired-end clean reads were merged as raw tags using FLASH [41] (v. 1.2.11) with a minimum overlap of 10 bp and mismatch error rates of 2%. Noisy sequences of raw tags were filtered by QIIME [42] (v. 1.9.1) pipeline under specific filtering conditions to obtain high-quality clean tags [43]. Reference-based chimera checking of clean tags was performed using the UCHIME algorithm (http://www.drive5.com/usearch/manual/uchime_algo.html) in the reference database (http://drive5.com/uchime/uchime_download.html). All chimeric tags were removed and effective tags for further analysis were obtained. The effective tags were clustered into OTUs greater than or equal to 97% similarity using the UPARSE pipeline. The tag sequence with the highest abundance was selected as the representative sequence within each cluster. Venn analysis was performed in R to identify unique and common OTUs among groups. The representative sequences were classified into organisms using a naive Bayesian model using RDP classifier [44] (Version 2.2) based on the Silva database [45]. The abundance statistics for each taxonomy and phylogenetic tree were constructed in a Perl script and visualized using SVG. Biomarker features for each group were screened using Metastats and LEfSe software. Weighted and unweighted UniFrac distance matrices were generated by QIIME. Multivariate statistical techniques using the non-metric multidimensional scaling (NMDS) of weighted UniFrac distances were calculated and plotted in R. The functional group (guild) of the OTUs was inferred using PICRUSt [46] (v. 1.1.0).

## 2.6. $CO_2$ emissions

$CO_2$ emissions were monitored using a static closed chamber, which was a little different to the method used by Li [47]. Gas samples were collected every 15 d (advanced or postponed if it rained) from static

chambers at check-points set on a ridge between two tobacco plants; three static chambers were placed in one plot. The dimensions of the static chamber were in accordance with Dyer [48]: the PVC pipe was 11 cm in diameter and 25 cm long and was placed 10 cm deep in the soil; a PVC cover of 11 cm was placed on the static chamber. A 20 cm diameter hole was made through the cover and secured with a rubber stopper. An airway (6 mm diameter, 10 cm long) near the hole was used to balance the pressure. Sampling was conducted between 09:00 and 11:00 under good weather conditions, and three complete gas measurements were obtained at three time points: $t = 0$, $t = 15$ and $t = 30$ min. A volume of 50 ml of injector was placed in the sampling window; before collection, the injector plunger was pushed three times to mix the gas in the static chamber. The injector was then placed in a hermetic bag and brought back to the laboratory when the gas collection was complete. Gas chromatography analysis (TRACEGC 2000, Italy) was used to check the sample. The $CO_2$ detector used was an electron capture detector (ECD); test temperature, 300°C; boiler temperature, 80°C; and carrier gas, pure $N_2$.

The accumulated emissions were calculated according to the following equation [49]:

$$\text{Accumulated } CO_2 \text{ emissions} = \sum_{i}^{n}(F_i \times D_i),$$

where $F_i$ is the rate of $CO_2$ emission flux (g m$^{-2}$ d$^{-1}$) in the $i_{th}$ sampling interval; $D_i$ is the number of days in the $i$th sampling interval and $n$ is the number of sampling intervals.

$CO_2$ emission flux (g m$^{-2}$ d$^{-1}$) was calculated according to the following equation [50]:

$$F = \left(\frac{V}{A}\right) \times \left(\frac{\Delta c}{\Delta t}\right) \times \left(\frac{273}{T}\right),$$

where $V$ is the volume of the chamber (m$^3$); $A$ is the area from which $CO_2$ emits into the chamber (m$^2$); $\Delta c / \Delta t$ = the rate of accumulation of $CO_2$ gas concentration in the chamber (mg m$^{-3}$ h$^{-1}$) and $T$ is the absolute temperature calculated as 273 plus mean temperature in the chamber (°C).

## 2.7. Calculation of the soil carbon pool management index

The carbon pool management index (CPMI) was calculated as follows.

Carbon pool activity (A) = soil labile organic carbon (mg g$^{-1}$)/soil unlabile organic carbon (mg g$^{-1}$)

Carbon pool activity index (AL) = A/reference soil carbon pool activity

Carbon pool index (CPI) = TC content (mg g$^{-1}$) of the sample/reference soil TC content (mg g$^{-1}$)

$$\text{Soil CPMI (\%)} = \text{CPI} \times \text{AL} \times 100.$$

## 2.8. Statistical analysis

Significant differences in soil characteristics were identified by one-way analysis of variance (ANOVA) followed by Duncan's-test using SPSS software (v. 17.0). NMDS ordination plots were used to identify differences in bacterial community composition, and redundancy analysis (RDA) was conducted to determine which environmental variables were most frequently related to the bacterial community. The NMDS and RDA analyses mentioned above were performed using the 'vegan' package in the R environment (R v. 3.2.0) [51]. Origin v. 9.0 was used to plot the column diagrams.

# 3. Results

## 3.1. Effects of biochar on the soil carbon pool

### 3.1.1. Effects of biochar on soil TC and total organic carbon

The TC content of purple and paddy soils increased after biochar was applied. In the paddy soil, after 1 year of biochar application, the TC content of the rhizosphere soil had not increased significantly compared to that of CK; however, after 2 years of biochar application, an improvement in the TC content of the rhizosphere soil was noted (figure 1). The TC content of the 0–20- and 20–40-cm soil layers improved in the first year and increased slightly after 2 years of biochar application. In the

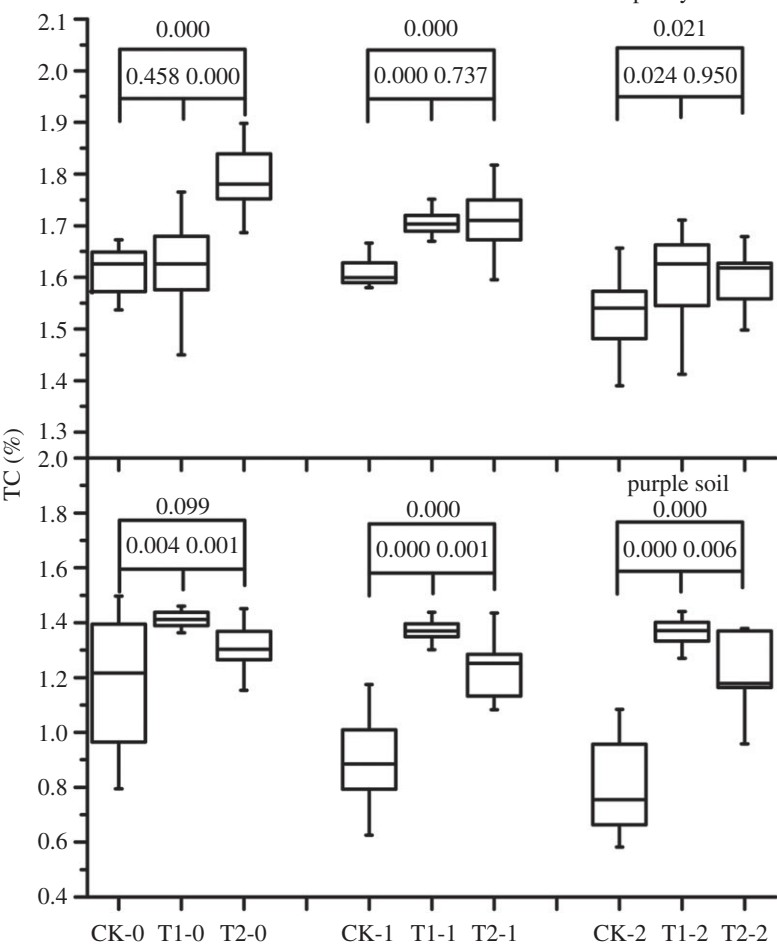

**Figure 1.** Effect of 2-year straw biochar application on the TC content of paddy and purple soils (CK, T1 and T2, respectively, refer to control, 1-year and 2-year biochar application; -0, -1 and -2, respectively, refer to rhizosphere soil and 0–20- and 20–40-cm soil layers; the numbers are *p* values; *n* = 15; this information is the same for figures 2–5).

purple soil, the TC content of each soil layer improved significantly after biochar application compared to that of CK soil; after 2 years of biochar application, a slight decline in TC was noted, but it was still significantly higher than that of CK.

Continuous application of biochar for 1 or 2 years was found to increase the TOC content in each soil layer (figure 2). The TOC content in the rhizosphere of purple and paddy soils increased after biochar application; this improvement was prominent in the 0–20- and 20–40-cm layers after 1 year of biochar application; 2 years of application decreased the TOC content, except in the 20–40-cm layer of the purple soil, but it was not lower than that of CK.

### 3.1.2. Effects of biochar on soil labile organic carbon content

After 2 years of biochar application, the DOC in the paddy and purple soils varied differently (figure 3). Compared with that in CK, the content of DOC in the rhizosphere, 0–20-cm, and 20–40-cm soil layers of the paddy soil, significantly increased with the duration of biochar application. Two years of biochar application increased the DOC content in the rhizosphere soil, 0–20-cm and 20–40-cm soil layers by 32.7%, 25.9% and 30.9%, respectively, compared with that in CK. In the purple soil, 2 years of consecutive biochar application had little effect on the DOC content.

The EOC content of purple and paddy soils (figure 4) increased significantly with increasing duration of biochar application. After 2 years of biochar application, the EOC in the rhizosphere, 0–20-cm and 20–40-cm soil layers increased by 7.8%, 17.8% and 22.1%, respectively, in the paddy soil and by 45.9%, 168.2% and 351.5%, respectively, in the purple soil.

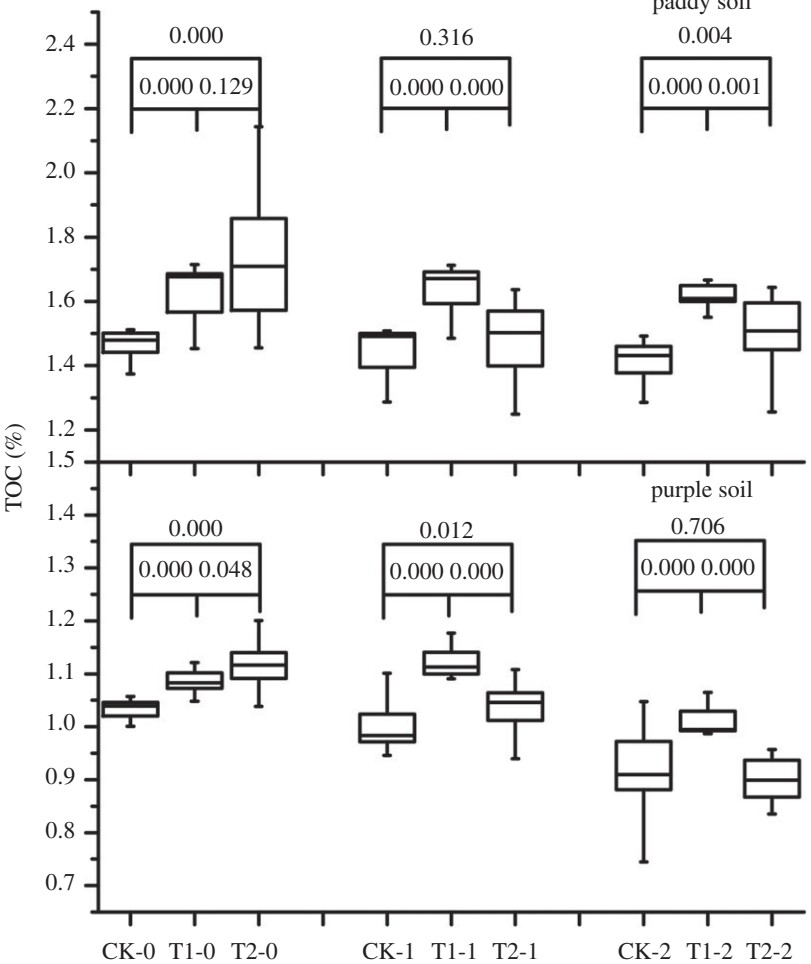

**Figure 2.** Effect of 2-year straw biochar application on the total organic carbon content in paddy and purple soils.

Biochar application significantly increased the soil MBC of the rhizosphere in purple and paddy soils (figure 5) by 5.8% and 24.7%, respectively, after 2 years of application. For the 0–20-cm and 20–40-cm soil layers, the effect of biochar application on MBC content was limited.

## 3.2. Effect of biochar on soil CPMI

The CPMI for the different treatments was calculated using the value for CK as a reference (table 2). Continuous application of biochar notably improved the carbon pool of the purple soil, and the A, AI and CPMI of the rhizosphere soil, 0–20-cm and 20–40-cm soil layers increased with an increase in the duration of biochar application. Two years of consecutive applications of biochar significantly affected soil carbon sequestration.

## 3.3. Effects of biochar on soil respiration

Continuous application of biochar significantly promoted $CO_2$ emissions in both paddy and purple soils (figure 6). In the paddy soil, compared with that of CK, the release of $CO_2$ increased by 23.25% and 29.16% for T1 and T2, respectively. The $CO_2$ release of purple soil was lower than that of paddy soil (12.95% and 32.36%, respectively, compared with that of CK). Thus, the straw biochar application played an important role in promoting $CO_2$ emissions.

## 3.4. Soil carbon pool and its activity are closely related to Lactobacillales and Bacteroidales abundance

For paddy soil and purple soil, 16S rDNA sequencing (five repeats) was carried out on soil samples from all treatments. A total of 2 230 977 tags were detected, and the number of tags per sample

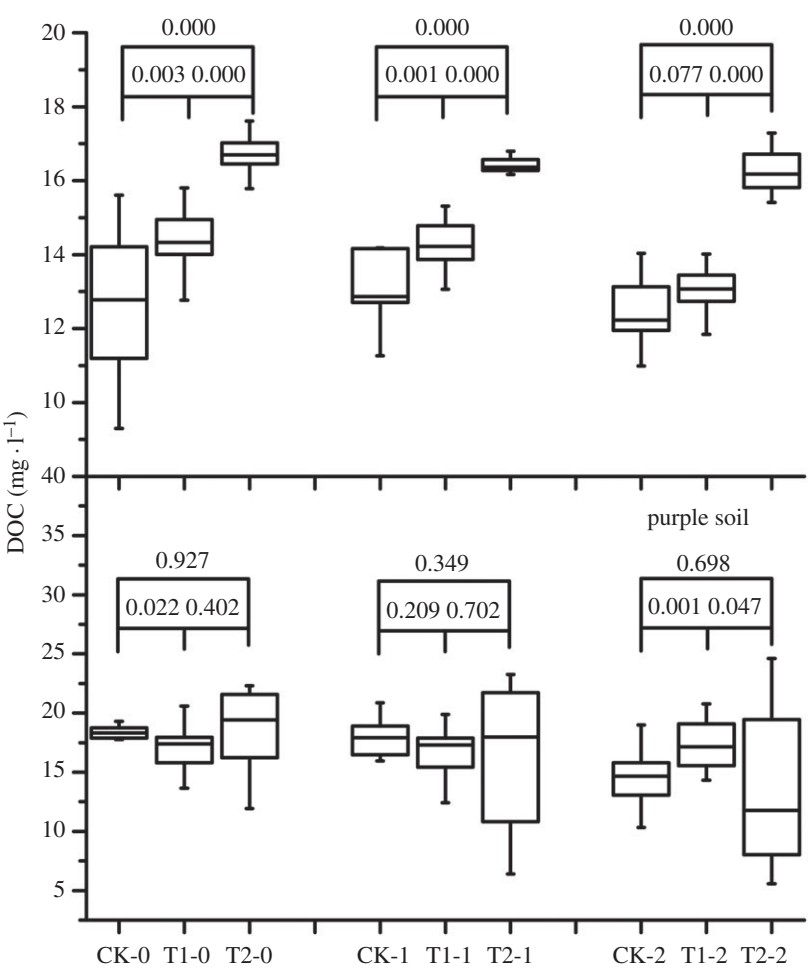

**Figure 3.** Effect of 2-year straw biochar application on the dissolved organic carbon content in paddy and purple soils.

ranged from 48 664 to 87 239. All effective tag sequences from all samples were clustered using UPARSE software; the sequences were clustered into OTUs with 97% consistency (identity), and the absolute abundance and relative information for each OTU in each sample were calculated (electronic supplementary material, figure S1). The proportion of the tags in the sample to the classification level of 'Family', 'Order' 'Class' and 'Phylum' is 72.47%, 69.28%, 59.82%, 42.31%, respectively. After 2 years, one of the samples was repeatedly noted to be a serious outlier, and it was excluded from subsequent analysis as an abnormal sample (electronic supplementary material, figure S2).

According to the species annotation and OTU abundance information, the functional annotation of KEGG pathways was performed using PICRUSt software [48], and the abundance information of each pathway and KO ID was counted. The results showed that the abundance of pathways in the process of carbon regeneration increased gradually with the treatment year, especially in purple soil. The results showed that changes in the microbial community were involved in the remediation of the soil carbon pool (electronic supplementary material, figure S3). For this reason, we further analysed the relationship between soil carbon pool components and soil microorganisms. At the order level, we totalled the abundances of the same microorganisms and calculated the total abundance for each microorganism, sorted them, and performed RDA for the microorganisms with abundances in the top 10. We found that the abundance of the bacterial order Lactobacillales was closely correlated with the TC, TOC, EOC, MBC, AI and CPMI of the soils and those of the bacterial orders Sphingomonadales and Rhizobiales were closely correlated with the DOC and A of the soils (figure 7).

Since a higher soil carbon index was found to be related to Lactobacillales, we focused on this order. We found that the abundance of Lactobacillales continued to increase during the continuous application of biochar in purple soil. Lactobacillales was 1.23% in purple soil and reached 3.76% and 4.40%, respectively, but the change in paddy soils was not obvious. In addition, we traced the Lactobacillales

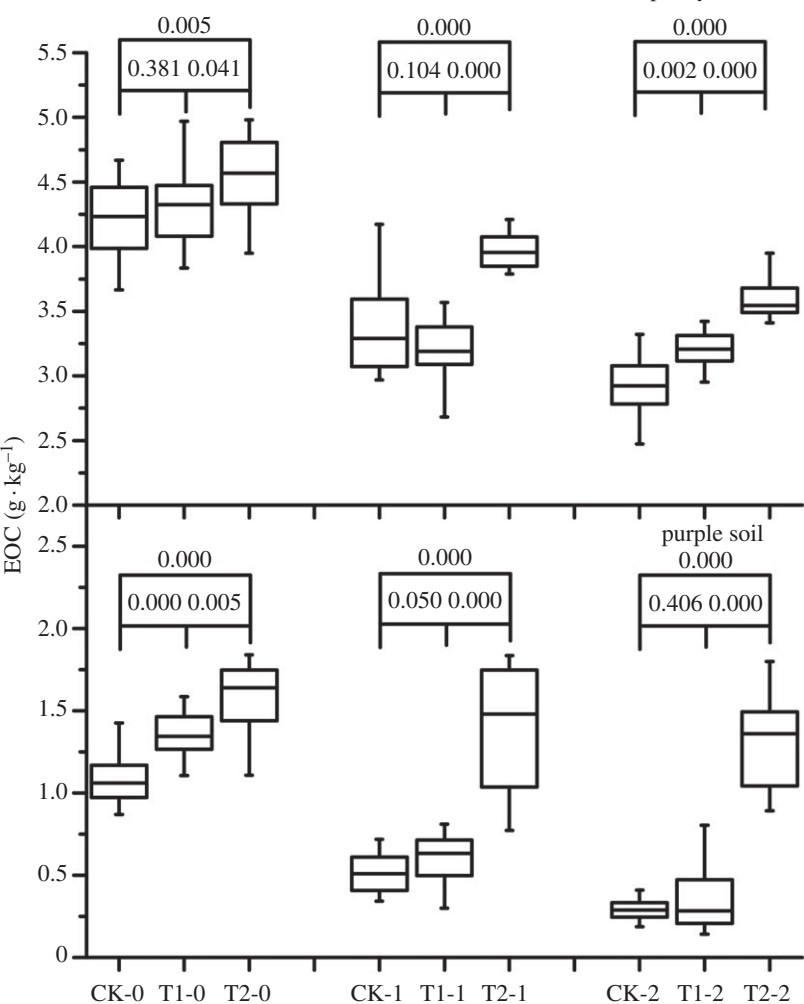

**Figure 4.** Effect of 2-year straw biochar application on the easily oxidized carbon content of paddy and purple soils.

order at higher and lower classification levels and found that the genus *Lactococcus* accounted for more than 90% of the microorganisms belonging to the order Lactobacillales, and the different treatments were associated with the level of the orders (Bacteria–Firmicutes–Bacillus–Lactobacillales–Streptococcaceae–Lactococcus). Thus, we inferred that the changes in Lactobacillales populations could be mainly attributed to the changes in the abundance of *Lactococcus* spp. (figure 8).

## 4. Discussion

In this study, application of biochar to two typical soils for 2 years enhanced the organic carbon content of the soil, which suggests that biochar affects the soil carbon pool. These results correspond to those of previous studies (table 3). This is mainly because on the one hand, SOM is mainly stabilized through complexation with Fe and Al oxyhydroxides and interaction with clay particles in subtropical soil, and biochar additions to soil enhance these organo-mineral interactions via adsorption and/or ligand exchange reactions, resulting in the stabilization and accumulation of SOC in soil [58–60]. On the other hand, the labile component leads to the addition of DOC to the soil carbon pool. Straw biochar contains numerous alkoxy and aryl groups; the proportion of aromatic carbon is around 70% [16,28]. Aromatic substances are known to have high biochemical stability [61,62]. The labile component of biochar mainly refers to the aliphatic and oxidizable carbon structure, which has heterogeneous chemical properties. The most labile component of biochar is the aliphatic compounds generated during the biochar preparation process [63]. This labile component of biochar can directly increase the SOC storage capacity in the solid form; aliphatic compounds gradually lose their peripheral structure and are released into the soil, directly decomposed by soil microbes and then converted to MBC and

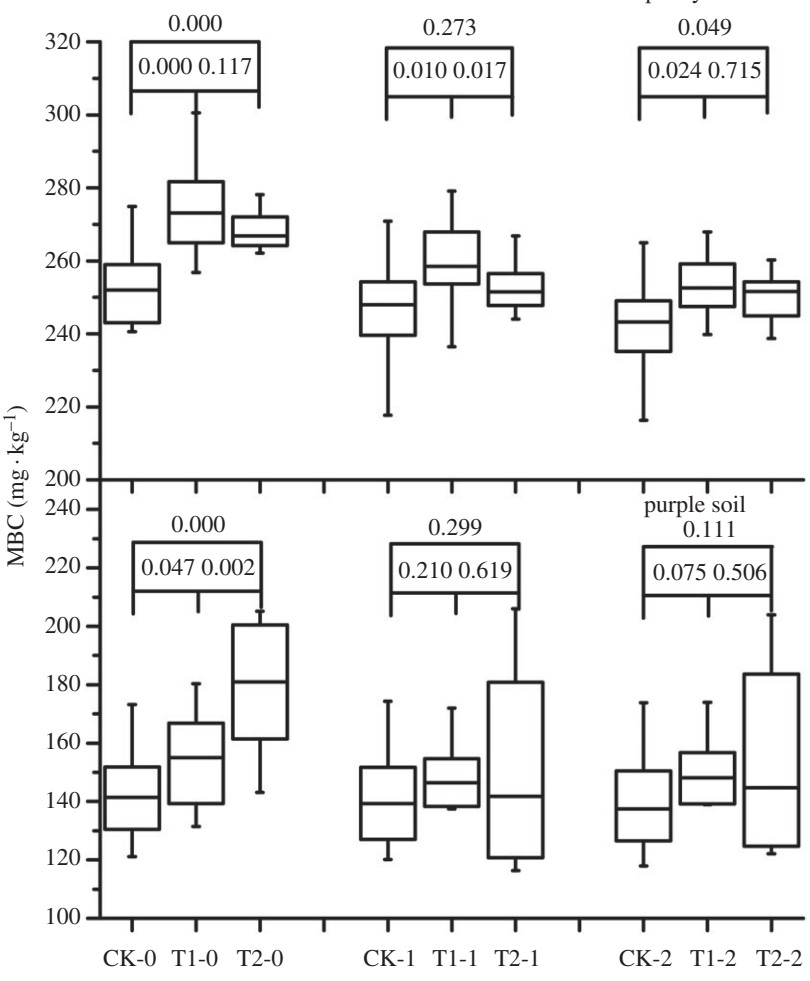

**Figure 5.** Effect of 2-year straw biochar application on the microbial biomass carbon content of paddy and purple soils.

**Table 2.** Effect of biochar on the soil carbon pool management index ($n = 15$). The lowercase letters indicate significant deference between different treatment ($p < 0.05$).

| treatment | | paddy soil | | | | purple soil | | | |
|---|---|---|---|---|---|---|---|---|---|
| | | A | AI | CPI | CPMI | A | AI | CPI | CPMI |
| rhizosphere | CK | 0.40a | 1.00a | 1.00a | 1.00b | 0.12a | 1.00c | 1.00a | 1.00b |
| | T1 | 0.36a | 0.89b | 1.01a | 0.90a | 0.14a | 1.22b | 1.19a | 1.45a |
| | T2 | 0.36a | 0.89b | 1.11a | 0.99b | 0.16a | 1.41a | 1.11a | 1.56a |
| 0–20 cm | CK | 0.30a | 1.00b | 1.00a | 1.00b | 0.05b | 1.00b | 1.00b | 1.00c |
| | T1 | 0.24a | 0.80c | 1.06a | 0.85c | 0.06b | 1.06b | 1.52a | 1.61b |
| | T2 | 0.36a | 1.21a | 1.06a | 1.29a | 0.15a | 2.82a | 1.37a | 3.87a |
| 20–40 cm | CK | 0.26a | 1.00a | 1.00a | 1.00b | 0.03b | 1.00b | 1.00b | 1.00c |
| | T1 | 0.25a | 0.95a | 1.05a | 1.00b | 0.03b | 1.03b | 1.71a | 1.76b |
| | T2 | 0.31a | 1.19a | 1.05a | 1.25a | 0.17a | 5.15a | 1.53a | 7.86a |

The lowercase letters indicate significant deference between different treatment (p<0.05).

DOM, thereby increasing the organic carbon pool of soil. Zhang [33] also reported that the addition of wheat biochar over two consecutive years during rice growth could significantly increase the DOC content from $0.82\,\mathrm{mg\,g^{-1}}$ and $0.35\,\mathrm{mg\,g^{-1}}$ to $0.94\,\mathrm{mg\,g^{-1}}$ and $0.53\,\mathrm{mg\,g^{-1}}$ in 2009 and 2010, respectively. Over time, microbial decomposition and transformation caused a reduction in the DOC

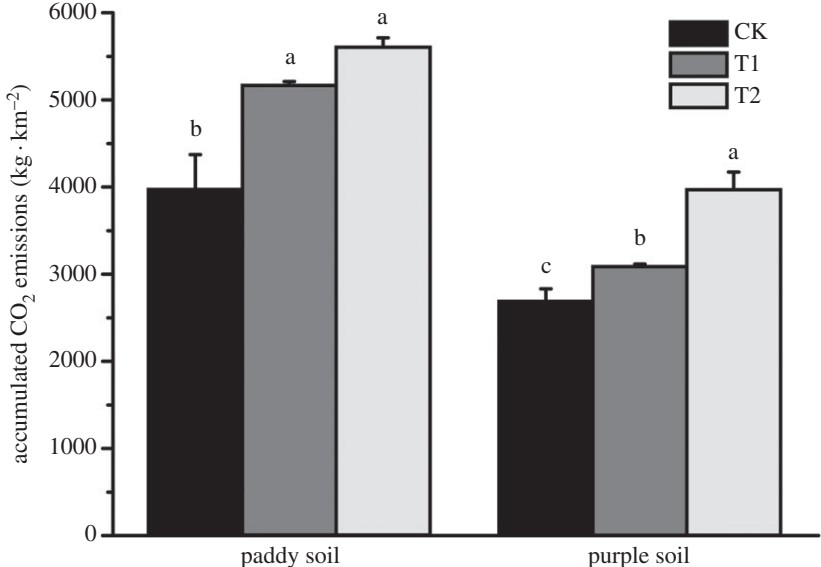

**Figure 6.** Two-year straw biochar application promoted soil respiration (different lowercase letters on the bars suggest a significant difference at $p < 0.05$).

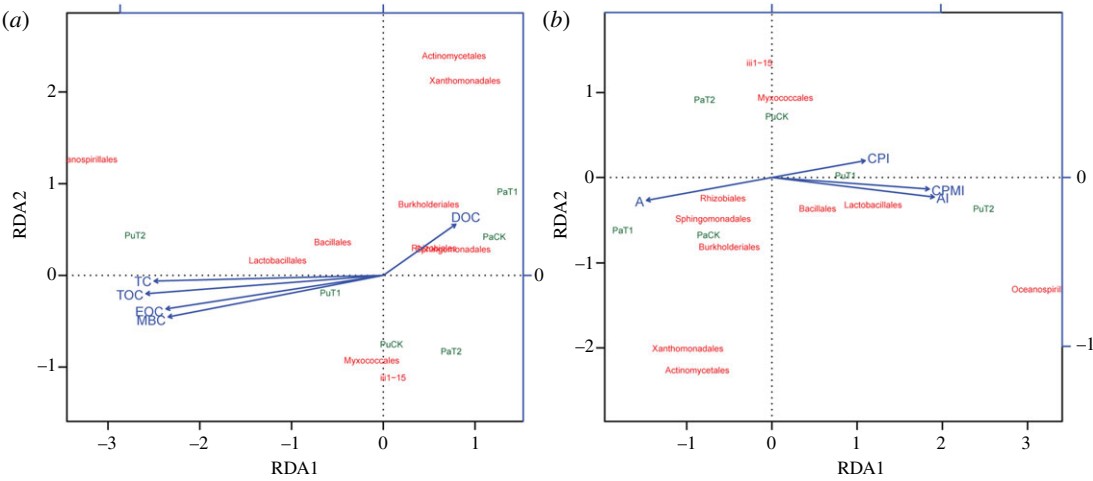

**Figure 7.** Redundancy analysis for microorganisms, soil samples, soil carbon pool and carbon pool, and the carbon pool management index ((a) is the RDA for microorganisms, soil sample and soil carbon pool; (b) is the analysis result for the microorganisms, soil sample and carbon pool management index; arrows represent the different environmental factors. Red and green texts represent different microbes and soil samples, respectively; the length of the arrows representing environmental factors conveys the degree of correlation between the corresponding environmental factors and research objects (samples or microorganisms), where the longer the arrow, the greater is the impact on the distribution of the object. The angle between the lines of each arrow represents the correlation, where an acute angle signifies a positive correlation between two environmental factors, and an obtuse angle signifies a negative correlation. The letters 'Pa' and 'Pu' refer to purple and paddy soils, respectively).

content; however, application of biochar maintained a high level of DOC compared to that in CK [64]. Therefore, biochar addition is a stable means of increasing the organic carbon pool.

Changes in soil microbial biomass are usually approximately represented by, and thus measured using, soil MBC [65,66]. In the present study, the addition of straw biochar significantly increased the MBC content of paddy and purple rhizosphere soils and improved the MBC content of the 0–20- and 20–40-cm soil layers. Thus, biochar can be added directly to the soil to supplement the SOC pool [67] and provide the most basic carbon source for the growth of soil microorganisms, promoting their reproduction and metabolism. Further, biochar leads to the deposition of N, P, K and other mineral nutrients, as well as a small amount of metal trace elements, promoting soil microbial reproduction [68,69].

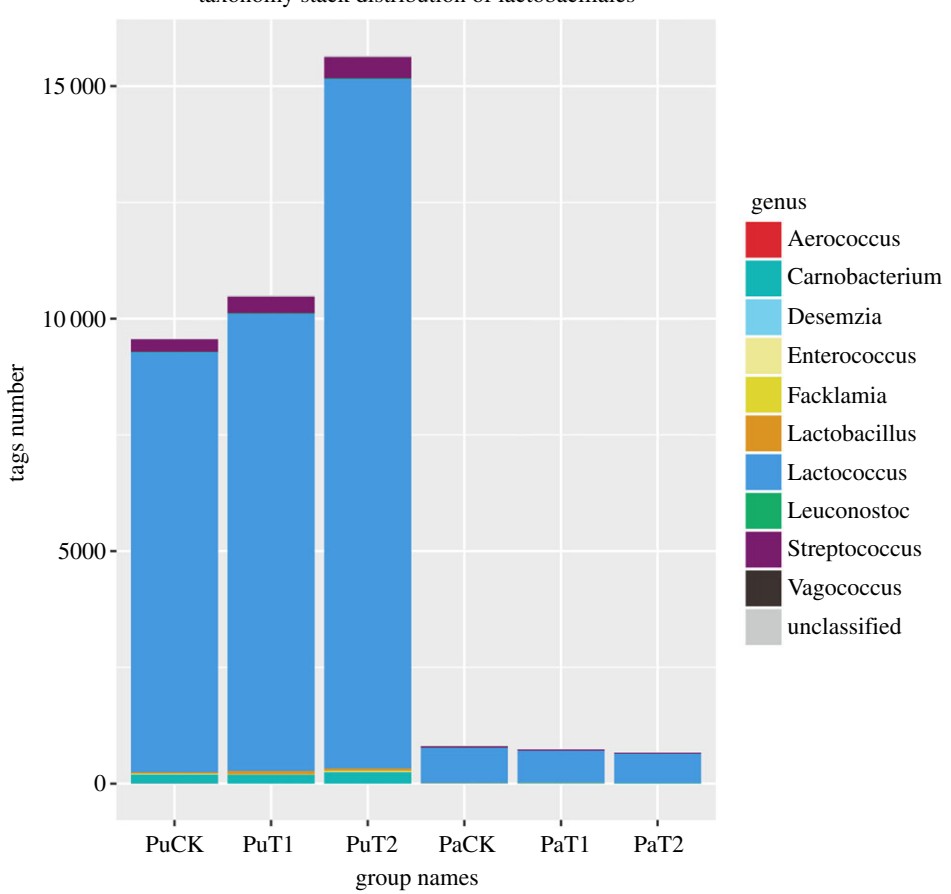

**Figure 8.** The abundance of Lactobacillales microorganisms in the soil increased after continuous application of microbial carbon from biochar.

Considering the carbon management index, the CPI of each layer of paddy and purple soils in southern China was remarkably affected by the consecutive application of biochar. Furthermore, application of biochar increased the TOC, stable carbon and CPI. In the purple soil, application of biochar slightly increased the A, AI, CPI and CPMI in the first year and increased them significantly in the second year. Thus, biochar application to purple soil affects carbon sequestration, which improved with increasing duration of biochar application. In paddy soils, biochar application decreased soil A and AI in the first year, indicating that biochar increased the organic carbon pool in the first year by increasing soil stable carbon content; however, A and AI significantly increased for each soil layer in the second year. These results showed that the SOC pool is increased by the increase in soil carbon content, eventually leading to an improvement in the activated carbon content. The effect of biochar on the soil carbon pool in paddy soil was, therefore, observed in the second production season.

Mineralization of soil organic matter leads to the conversion of organic compounds to inorganic ones, leading to the release of mineral nutrient ions under the action of soil microorganisms. We found that the consecutive application of biochar over 2 years resulted in increased soil $CO_2$ emissions in both soil types. This result could be attributed to the efficient degradation of organic matter by biochar, which in turn promoted growth and metabolism of soil microbes, stimulating degradation of soil organic matter rather than fixing it to reach a stable state [70,71]. In addition, biochar addition has been shown to improve the mineralization of soil organic matter by changing the physical and chemical properties of soil, such as increasing temperature, soil porosity, pH, water-holding capacity, nutrient uptake and oxygen content [72–75]. Contrary to this finding, some studies have shown that application of biochar adversely affects the mineralization of soil organic matter [76–78]. The discrepancy in this effect could be attributed to the type of biomass carbon, production conditions and climatic factors.

After biochar application in purple soil, the increase in specific microorganisms related to MBC and soil carbon was significantly higher than that in paddy soil. The changes in soil microbial community

**Table 3.** Comparison of the effect of biochar on both changes in soil carbon, microbial communities and $CO_2$ in this study with other reports.

| place | biochar type | process | soil type | soil organic carbon | soil bacteria | $CO_2$ emission | reference |
|---|---|---|---|---|---|---|---|
| Guanghan, Sichuan province | wheat straw | pyrolysis between 350 and 550°C | aquept | ↑ | ↑ | — | [52] |
| Jinxian, Jiangxi province | wheat straw | pyrolysis between 350 and 550°C | paddy soils | ↑ | ↑ | — | [53] |
| Changsha Hunan Province | wheat straw | pyrolysis between 350 and 550°C | paddy soils | ↑ | ↑ | — | [53] |
| Guanghan Sichuan province | wheat straw | pyrolysis between 350 and 550°C | paddy soils | ↑ | ↑ | — | [53] |
| Guanghan, Sichuan province | wheat straw | pyrolysis between 350 and 550°C | hydragric anthrosol | ↑ | ↑ | — | [54] |
| Lin'an City, Zhejiang province | bamboo leaf | 500°C under an oxygen-limited condition | — | — | — | ↑ | [55] |
| Yixing, Jiangsu Province | wheat straw | pyrolysis between 350 and 550°C | paddy soil | — | — | no effect | [56] |
| Shangqiu City, Henan Province | wheat straw | pyrolysis between 350 and 550 °C | aquic fluvent | — | — | ↑[a] | [57] |
| Xinfeng, Jiangxi province | Rice straw | pyrolysis between 350 and 550°C | paddy soils | ↑ | ↑ | ↑ | this study |
| Xinfeng, Jiangxi province | Rice straw | pyrolysis between 350 and 550°C | purple soil | ↑ | ↑ | ↑ | this study |

[a]Biochar increased the total $CO_2$ emission without N fertilization application.

abundance varied between the two types of soil, which could mainly be attributed to differences in soil texture, pH and soil C:N ratio. In the present study, application of straw biochar promoted the microbial abundance of Lactobacillales, and the changes in Lactobacillales abundance were mainly caused by changes in the abundance of *Lactococcus* spp. *Lactococcus* is believed to contribute to soil nitrogen carbon cycling [79], potentially impacts on carbon sequestration [80], affects soil fertility and helps improve crop growth [81]. This organism has a high heterotrophic metabolism and plays an important role in the decomposition of SOC [82]. A previous study showed that *Lactococcus* spp. abundance was positively correlated with soil respiration [83]. In addition, the microbial activity of anaerobic bacteria was lower than that of aerobic bacteria and facultative aerobic bacteria. Thus, *Lactococcus* spp. should have higher respiration and the observed changes in abundance may be related to the increase in soil respiration after biochar application.

Overall, in this study, application of biochar significantly improved soil composition, microbial abundance and soil respiration, especially in purple soils. These results confirmed the improvement effect of biochar on soil carbon pools in southern China, which is of great significance for guiding soil improvement in production systems.

# 5. Conclusion

Taken together, the results of this study showed that biochar application in purple and paddy soils could increase the level and activity of the soil carbon pool, and thus soil respiration. The soil carbon pool was found to be closely related to the abundance of Lactobacillales, and *Lactococcus* spp. were important species accounting for the improvement in the soil carbon pool. These results provide a theoretical basis for conducting further studies to determine the relationship between biochar application and soil quality and soil microecology in subtropical and tropical regions. In the next stage, we will continue to study how biochar affects soil carbon pool components to further deepen our understanding of the mechanisms of biochar improvement in soil carbon pools.

Data accessibility. The data that support the findings of this study are openly available from the Dryad Digital Repository at: https://doi.org/10.5061/dryad.m5j1rs3 [84].

Authors' contributions. S.Y. and H.Z. carried out the molecular laboratory work, participated in data analysis, carried out sequence alignments, participated in the design of the study and drafted the manuscript; H.Y. carried out the statistical analyses; S.Y., Z.N. and A.Z. collected field data; S.Y., K.H., X.X., N.W., C.G. and G.L. conceived of the study, designed the study, coordinated the study and helped draft the manuscript. All authors gave final approval for publication.

Competing interests. The authors declare no conflict of interest.

Funding. This research did not receive any specific grant from funding agencies in the public, commercial, or not-for-profit sectors.

Acknowledgements. The authors thank Yi Liu, Wenping Rao, Fangfang Han, Yun Gao and Chang Ma for their assistance in obtaining field and crop measurements.

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
