## [Reviewer comments · Royal Society Open Science]

Review History

RSOS-181499.R0 (Original submission)

Review form: Reviewer 1

Is the manuscript scientifically sound in its present form?

No

Are the interpretations and conclusions justified by the results?

Yes

Is the language acceptable?

Yes

Is it clear how to access all supporting data?

No

Do you have any ethical concerns with this paper?

No

Have you any concerns about statistical analyses in this paper?

No

Recommendation?

Major revision is needed (please make suggestions in comments)

Comments to the Author(s)

This is a well written paper with a sound experimental design.

There have been many biochar studies done in paddy soils in China examining both changes in soil carbon and microbial communities. Please make a table to show what studies have been done and whether they have found different results to that given in this study. In particular there have been many studies published using San Li wheat straw biochar. (Joseph et al 2013, Qian Li et al 2014 etc)

line 103 You must specify what type of biochar and temperature it was made at. Generalisations dont help the reader.

Section 2.2 Please give details of the properties of the biochars , feedstock used and temperature of manufacture and device used to make the biochar t the beginning of this section. There is not enough data in line 152

Line 228 Please state how your method differs from other methods where GHG emissions were measured (e.g. Qian Li 57. Qian L., (2014). Biochar compound fertilizer as an option to reach high productivity but low carbon intensity in rice agriculture of China. Carbon Management. 5(2):pp 145-158

3.1.1 I would like to see the data graphed to show the increase in no biochar C that has built up in the soil. Please see 1. Weng Z., Van Zwieten L., ., 2017 Biochar builds soil carbon over a decade by stabilising rhizodeposits; Nature Climate Change; 7, (5) 371-376

The aromatic carbon in wheat straw biochar is not 98.5 % Please quote the figures in Joseph et al 2013 Carbon Management Shifting Paradigms

You need to give a more detailed explanation of the role of Lactococcus in the increase in yield of rice or in the changes in carbon pools

Review form: Reviewer 2**Is the manuscript scientifically sound in its present form?**

Yes

Are the interpretations and conclusions justified by the results?

Yes

Is the language acceptable?

Yes

Is it clear how to access all supporting data?

Yes

Do you have any ethical concerns with this paper?

No

Have you any concerns about statistical analyses in this paper?

Yes

Recommendation?

Accept with minor revision (please list in comments)

Comments to the Author(s)

The manuscript deals with current issue of biochar application to soil to redistribute CO₂ and improve soil carbon pool. The paper is very well written and the results are novel and valuable since the utilization of agricultural waste materials such as straw may be a step forward in protection of natural environment and simultaneous improvement of cultivation soils. However, the manuscript needs minor corrections before the acceptance:

- 1) Data presented in tables should be supported by statistical analysis and the "n" value should be presented in table legends and figure captions,
- 2) The Introduction is lengthy and should be definitely shortened.

Decision letter (RSOS-181499.R0)

07-Feb-2019

Dear Professor Liu,

The editors assigned to your paper ("Biochar application to paddy and purple soils in Southern China: alternation of soil carbon and biotic activity") have now received comments from reviewers. We would like you to revise your paper in accordance with the referee and Associate Editor suggestions which can be found below (not including confidential reports to the Editor). Please note this decision does not guarantee eventual acceptance.

Please submit a copy of your revised paper before 02-Mar-2019. Please note that the revision deadline will expire at 00.00am on this date. If we do not hear from you within this time then it will be assumed that the paper has been withdrawn. In exceptional circumstances, extensions may be possible if agreed with the Editorial Office in advance. We do not allow multiple rounds of revision so we urge you to make every effort to fully address all of the comments at this stage. If deemed necessary by the Editors, your manuscript will be sent back to one or more of the original reviewers for assessment. If the original reviewers are not available, we may invite new reviewers.

- Data accessibility

If you wish to submit your supporting data or code to Dryad (<http://datadryad.org/>), or modify your current submission to dryad, please use the following link:
<http://datadryad.org/submit?journalID=RSOS&manu=RSOS-181499>

- Competing interests

- Authors' contributions

- Acknowledgements

- Funding statement

on behalf of Dr Berat Haznedaroglu (Associate Editor) and Professor Kevin Padian (Subject Editor)
 openscience@royalsociety.org

Comments to Author:
 Reviewers' Comments to Author:
 Reviewer: 1

Comments to the Author(s)
 This is a well written paper with a sound experimental design.

There have been many biochar studies done in paddy soils in China examining both changes in soil carbon and microbial communities. Please make a table to show what studies have been done and whether they have found different results to that given in this study. In particular there have been many studies published using San Li wheat straw biochar. (Joseph et al 2013, Qian Li et al 2014 etc)

line 103 You must specify what type of biochar and temperature it was made at. Generalisations dont help the reader.

Section 2.2 Please give details of the properties of the biochars , feedstock used and temperature of manufacture and device used to make the biochar t the beginning of this section. There is not enough data in line 152

Line 228 Please state how your method differs from other methods where GHG emissions were measured (e.g. Qian Li 57. Qian L., (2014). Biochar compound fertilizer as an option to reach high productivity but low carbon intensity in rice agriculture of China. Carbon Management. 5(2):pp 145-158

3.1.1 I would like to see the data graphed to show the increase in no biochar C that has built up in the soil. Please see 1. Weng Z., Van Zwieten L., ., 2017 Biochar builds soil carbon over a decade by stabilising rhizodeposits; Nature Climate Change; 7, (5) 371-376

The aromatic carbon in wheat straw biochar is not 98.5 % Please quote the figures in Joseph et al 2013 Carbon Managment Shifting Paradigms

You need to give a more detailed explanation of the role of Lactococcus in the increase in yield of rice or in the changes in carbon pools

Reviewer: 2

Comments to the Author(s)
 The manuscript deals with current issue of biochar application to soil to redistribute CO2 and improve soil carbon pool. The paper is very well written and the results are novel and valuable

since the utilization of agricultural waste materials such as straw may be a step forward in protection of natural environment and simultaneous improvement of cultivation soils. However, the manuscript needs minor corrections before the acceptance:

- 1) Data presented in tables should be supported by statistical analysis and the “n” value should be presented in table legends and figure captions,
- 2) The Introduction is lengthy and should be definitely shortened.

Author's Response to Decision Letter for (RSOS-181499.R0)

See Appendix A.

RSOS-181499.R1 (Revision)

Review form: Reviewer 1

Is the manuscript scientifically sound in its present form?

No

Are the interpretations and conclusions justified by the results?

No

Is the language acceptable?

Yes

Is it clear how to access all supporting data?

No

Do you have any ethical concerns with this paper?

No

Have you any concerns about statistical analyses in this paper?

No

Recommendation?

Major revision is needed (please make suggestions in comments)

Comments to the Author(s)

line 82 what is dry combustion. 760C is a very high temperature for biochar. you need to discuss the role of temperature in changes in soil properties and microbial biomass. Suggest you start with Joseph et al Shifting paradigms Carbon Management.

You state that few studies have been done Please list all of these studies. Parts of China has a long history of making and using straw biochar please correct the statement.

There have been many groups who have done greenhouse gas measurements using straw biochar in China. Please list these. What is novel. Has anyone done studies with tobacco. I have

measured the maximum temperature in San Li reactor at approximately 480C. This has been reported in Joseph et al 2013. Please alter and add the reference.
 How does your techniques differ from those used by say Genxing Pans Group who have done extensive greenhouse gas measurement with San Li wheat straw.

Please clarify whether the carbon increases were just due to the addition of the biochar or to the addition of biochar and fixation. I refer the authors to the study by Weng et al Nature Climate Change; 7, (5) 371-376

So if total C increased in the soil why did CO₂ increase. Where is the extra carbon coming from? The discussion does not really include the most recent reports on how biochar builds up stable organo mineral layers. I suggest the authors read Archanjo et al 2(2017) Nanoscale analyses of the surface structure and composition of biochars extracted from field trials or after co-composting using advanced analytical electron microscopy. Geoderma . 295,70-79 . Hagemann ,et al. (2017) Organic coating on biochar explains its nutrient retention and stimulation of soil fertility. Nature Communications 18, 1089 and Weng et al

Who else has found that Lactococcus plays a major role in soil carbon pool and what other bacteria have other researchers found that increase carbon when biochar is added?

Review form: Reviewer 2

Is the manuscript scientifically sound in its present form?

Yes

Are the interpretations and conclusions justified by the results?

Yes

Is the language acceptable?

Yes

Is it clear how to access all supporting data?

Yes

Do you have any ethical concerns with this paper?

No

Have you any concerns about statistical analyses in this paper?

No

Recommendation?

Accept as is

Comments to the Author(s)

The manuscript has been adjusted according to the reviewers' comments and in my opinion can be published in a present form.

Decision letter (RSOS-181499.R1)

25-Mar-2019

Dear Professor Liu:

Manuscript ID RSOS-181499.R1 entitled "Biochar application on paddy and purple soils in southern China: soil carbon and biotic activity" which you submitted to Royal Society Open Science, has been reviewed. The comments of the reviewer(s) are included at the bottom of this letter.

Please submit a copy of your revised paper before 17-Apr-2019. Please note that the revision deadline will expire at 00.00am on this date. If we do not hear from you within this time then it will be assumed that the paper has been withdrawn. In exceptional circumstances, extensions may be possible if agreed with the Editorial Office in advance. We do not allow multiple rounds of revision so we urge you to make every effort to fully address all of the comments at this stage. If deemed necessary by the Editors, your manuscript will be sent back to one or more of the original reviewers for assessment. If the original reviewers are not available we may invite new reviewers.

- Ethics statement

- Data accessibility

- Competing interests

- Authors' contributions

- Acknowledgements

- Funding statement

Kind regards,

Royal Society Open Science Editorial Office
Royal Society Open Science
openscience@royalsociety.org

on behalf of Dr Berat Haznedaroglu (Associate Editor) and Professor Kevin Padian (Subject Editor)
openscience@royalsociety.org

Reviewer comments to Author:

Reviewer: 1

Comments to the Author(s)

line 82 what is dry combustion. 760C is a very high temperature for biochar. you need to discuss the role of temperature in changes in soil properties and microbial biomass. Suggest you start with Joseph et al Shifting paradigms Carbon Management.

You state that few studies have been done Please list all of these studies. Parts of China has a long history of making and using straw biochar please correct the statement.

There have been many groups who have done greenhouse gas measurements using straw

biochar in China. Please list these. What is novel. Has anyone done studies with tobacco. I have measured the maximum temperature in San Li reactor at approximately 480C. This has been reported in Joseph et al 2013. Please alter and add the reference.

How does your techniques differ from those used by say Genxing Pans Group who have done extensive greenhouse gas measurement with San Li wheat straw.

Please clarify whether the carbon increases were just due to the addition of the biochar or to the addition of biochar and fixation. I refer the authors to the study by Weng et al Nature Climate Change; 7, (5) 371-376

So if total C increased in the soil why did CO₂ increase. Where is the extra carbon coming from? The discussion does not really include the most recent reports on how biochar builds up stable organo mineral layers. I suggest the authors read Archanjo et al 2(2017) Nanoscale analyses of the surface structure and composition of biochars extracted from field trials or after co-composting using advanced analytical electron microscopy. Geoderma . 295,70-79 . Hagemann ,et al. (2017) Organic coating on biochar explains its nutrient retention and stimulation of soil fertility. Nature Communications 18, 1089 and Weng et al

Who else has found that Lactococcus plays a major role in soil carbon pool and what other bacteria have other researchers found that increase carbon when biochar is added?

Reviewer: 2

Comments to the Author(s)

The manuscript has been adjusted according to the reviewers' comments and in my opinion can be published in a present form.

Author's Response to Decision Letter for (RSOS-181499.R1)

See Appendix B.

RSOS-181499.R2 (Revision)

Review form: Reviewer 1

Is the manuscript scientifically sound in its present form?

Yes

Are the interpretations and conclusions justified by the results?

Yes

Is the language acceptable?

Yes

Is it clear how to access all supporting data?

Yes

Do you have any ethical concerns with this paper?

No

Have you any concerns about statistical analyses in this paper?

I do not feel qualified to assess the statistics

Recommendation?

Accept with minor revision (please list in comments)

Comments to the Author(s)

Much improved paper. Further editing of the english would make the paper more readable. They refer to the non aromatic carbon as unstable. This is not correct as some of the non aromatic warer soluble organic compounds have similar structures to humic acids and polyphenols. I would recommend using the term labile

Review form: Reviewer 2

Is the manuscript scientifically sound in its present form?

Yes

Are the interpretations and conclusions justified by the results?

Yes

Is the language acceptable?

Yes

Is it clear how to access all supporting data?

Yes

Do you have any ethical concerns with this paper?

No

Have you any concerns about statistical analyses in this paper?

No

Recommendation?

Accept as is

Comments to the Author(s)

This is the second revision of this manuscript, and it has been improved in both readability and scientific clarity. I have no additional comments and I recommend the manuscript for publication.

Decision letter (RSOS-181499.R2)

21-May-2019

Dear Professor Liu:

On behalf of the Editors, I am pleased to inform you that your Manuscript RSOS-181499.R2 entitled "Biochar application on paddy and purple soils in southern China: soil carbon and biotic activity" has been accepted for publication in Royal Society Open Science subject to minor revision in accordance with the referee suggestions. Please find the referees' comments at the end of this email.

The reviewers and Subject Editor have recommended publication, but also suggest some minor revisions to your manuscript. Therefore, I invite you to respond to the comments and revise your manuscript.

- Ethics statement

- Data accessibility

If you wish to submit your supporting data or code to Dryad (<http://datadryad.org/>), or modify your current submission to dryad, please use the following link:
<http://datadryad.org/submit?journalID=RSOS&manu=RSOS-181499.R2>

- Competing interests

- Authors' contributions

- Acknowledgements

- Funding statement

Because the schedule for publication is very tight, it is a condition of publication that you submit the revised version of your manuscript before 30-May-2019. Please note that the revision deadline will expire at 00.00am on this date. If you do not think you will be able to meet this date please let me know immediately.

Supplementary files will be published alongside the paper on the journal website and posted on

the online figshare repository (<https://figshare.com>). The heading and legend provided for each supplementary file during the submission process will be used to create the figshare page, so please ensure these are accurate and informative so that your files can be found in searches. Files on figshare will be made available approximately one week before the accompanying article so that the supplementary material can be attributed a unique DOI.

on behalf of Dr Berat Haznedaroglu (Associate Editor) and Kevin Padian (Subject Editor)
openscience@royalsociety.org

Reviewer comments to Author:

Reviewer: 1

Comments to the Author(s)

Much improved paper. Further editing of the english would make the paper more readable. They refer to the non aromatic carbon as unstable. This is not correct as some of the non aromatic warer soluble organic compounds have similar structures to humic acids and polyphenols. I would recommend using the term labile

Reviewer: 2

Comments to the Author(s)

This is the second revision of this manuscript, and it has been improved in both readability and scientific clarity. I have no additional comments and I recommend the manuscript for publication.

Author's Response to Decision Letter for (RSOS-181499.R2)

See Appendix C.

Decision letter (RSOS-181499.R3)

31-May-2019

Dear Professor Liu,

I am pleased to inform you that your manuscript entitled "Biochar application on paddy and

purple soils in southern China: soil carbon and biotic activity" is now accepted for publication in Royal Society Open Science.

on behalf of Dr Berat Haznedaroglu (Associate Editor) and Kevin Padian (Subject Editor)
openscience@royalsociety.org

Follow Royal Society Publishing on Twitter: [@RSocPublishing](https://twitter.com/RSocPublishing)
Follow Royal Society Publishing on Facebook:
<https://www.facebook.com/RoyalSocietyPublishing.FanPage/>
Read Royal Society Publishing's blog: <https://blogs.royalsociety.org/publishing/>

Appendix A

Journal: *Royal Society Open Science*

Manuscript ID: RSOS-181499

Manuscript Title: Biochar application on paddy and purple soils in southern China: soil carbon and biotic activity

Author(s): Shen Yan, Zhengyang Niu, Aigai Zhang, Haitao Yan, He Zhang, Kuanxin He, Xianyi Xiao, Nianlei Wang, Chengwei Guan, Guoshun Liu

Dear editor and reviewers,

Thank you very much for giving us the opportunity to revise and resubmit our manuscript. We highly appreciated the professional and illuminating comments provided by you. These comments and suggestions were encouraging and helpful. We have tried our best to address the points and issue carefully and revise the manuscript accordingly. We provide detailed, point-by-point responses in the following pages. Note that the reviewer' comments are presented in black font, and our responses are in blue font. All changed made in the revised manuscript are marked using yellow highlight. Please let me know if you have any further questions and suggestions. We hope the revision will be satisfactory.

Sincerely,

Shen Yan and Guoshun Liu

Reviewer #1:

Comment 1: There have been many biochar studies done in paddy soils in China examining both changes in soil carbon and microbial communities. Please make a table to show what studies have been done and whether they have found different results to that given in this study. In particular there have been many studies published using San Li wheat straw biochar. (Joseph et al 2013, Qian Li et al 2014 etc)

Response: Thanks for your review and helpful suggestion. We have already made the table and put it into the discussion part (Lines 333;428-429).

Comment 2: line 103 You must specify what type of biochar and temperature it was made at. Generalisations don't help the reader.

Response: Thank you for the comments. The type of biochar and temperature was specific in current manuscript (Lines 82-83).

Comment 3: Section 2.2 Please give details of the properties of the biochars, feedstock used and temperature of manufacture and device used to make the biochar the beginning of this section. There is not enough data in line 152

Response: Thanks for your review. The details of the properties of the biochar, feedstock used and temperature of manufacture and device

used were added in the current manuscript (Lines 129-131), and we moved the details the properties of the biochar to the Table S1.

Comment 4: Line 228 Please state how your method differs from other methods where GHG emissions were measured (e.g. Qian Li 57. Qian L., (2014). Biochar compound fertilizer as an option to reach high productivity but low carbon intensity in rice agriculture of China. Carbon Management. 5(2):pp 145-158

Response: Thanks for your review. These two methods are kind of same, but they still have some different. We use different size of static chamber, what is more, our gas sampling was done at 15d interval while for Qian' s method the interval was 7d. these changes will reflect on the formula calculation in our paper.

Comment 5: 3.1.1 I would like to see the data graphed to show the increase in no biochar C that has built up in the soil. Please see 1.

Weng Z., Van Zwieten L., ., 2017 Biochar builds soil carbon over a decade by stabilising rhizodeposits; Nature Climate Change; 7, (5) 371-376

Response: Thanks for your review. This paper is helpful for our research, however, we did not test the data for no biochar C built up in the soil, which need to use C13 to achieve it. As a helpful suggestion, we will adopt this method in our further study.

Comment 6: The aromatic carbon in wheat straw biochar is not 98.5 %
Please quote the figures in Joseph et al 2013 Carbon Management
Shifting Paradigms.

Response: Thanks for your review. This reference is very useful, and I
have already added it to revised manuscript (Lines 336-337).

Comment 7: You need to give a more detailed explanation of the role
of Lactococcus in the increase in yield of rice or in the changes in
carbon pools

Response: Thanks for your review and valuable comments. We have
revised this section according to your suggestion. And the detail of the
role of Lactococcus were added in the current manuscript (Lines 394-
398)

Reviewer #2:

Comment 1: Data presented in tables should be supported by
statistical analysis and the “n” value should be presented in table
legends and figure captions.

Response: Thanks for your helpful suggestions. We have already
change the table 2 and put “n” value in table legends and figure
captions(Lines 424;437).

Comment 2: The Introduction is lengthy and should be definitely shortened.

Response: Thank you for your review and valuable advices. We have revised this section according to your suggestion. The introduction has been shortened, and these changes will not influence the content and framework of the paper.

Appendix B

Journal: *Royal Society Open Science*

Manuscript ID: RSOS-181499

Manuscript Title: Biochar application on paddy and purple soils in southern China: soil carbon and biotic activity

Author(s): Shen Yan, Zhengyang Niu, Aigai Zhang, Haitao Yan, He Zhang, Kuanxin He, Xianyi Xiao, Nianlei Wang, Chengwei Guan, Guoshun Liu

Dear editor and reviewers,

Thank you very much for giving us the opportunity to revise and resubmit our manuscript. We highly appreciated the professional and illuminating comments provided by you. These comments and suggestions were encouraging and helpful. We have tried our best to address the points and issues carefully and revised the manuscript accordingly. We provide detailed, point-by-point responses in the following pages. Note that the reviewer' comments are presented in black font, and our responses are in blue font. All changed made in the revised manuscript are marked using yellow highlight. Please let me know if you have any further questions and suggestions. We hope the revision will be satisfactory.

Sincerely,

Shen Yan and Guoshun Liu

Reviewer #1:

1 line 82 what is dry combustion. 760C is a very high temperature for biochar. you need to discuss the role of temperature in changes in soil properties and microbial biomass. Suggest you start with Joseph et al Shifting paradigms Carbon Management.

Thanks for your review and helpful suggestion. We made a mistake and the temperature is 700 °C, also, we delete the “dry combustion” which was also used by mistake. However, 700 °C is also very high for the biochar, thus we discussed the role of temperature for biochar in changes in soil properties and microbial biomass, and the reference you suggested is very helpful and we have added this reference into our manuscript. (line 82-87)

2 You state that few studies have been done Please list all of these studies. Parts of China has a long history of making and using straw biochar please correct the statement.

Thanks for your review, we have already listed the studies and corrected the statement about the straw biochar making and using in the history of China. (line 90-93,94-96)

3 There have been many groups who have done greenhouse gas measurements using straw biochar in China. Please list these. What is novel. Has anyone done studies with tobacco.

Thanks for your review. There is no study on greenhouse gas in tobacco field, and we add this statement in the manuscript (line 95-96) . Also, we add the studies using straw biochar in China in table 3.

I have measured the maximum temperature in San Li reactor at approximately 480C. This has been reported in Joseph et al 2013. Please alter and add the reference.

Thanks for your review. We have added this reference into the manuscript (line 133-134) .

4 How does your techniques differ from those used by say Genxing Pans Group who have done extensive greenhouse gas measurement with San Li wheat straw.

Thanks for your review. we have made this statement in the paper.

These two methods are kind of same, but we still have some differences. We use different size of static chamber, what is more, our gas sampling was done at 15d interval while for Li' s method the interval was 7d. these changes will reflect on the formula calculation in our paper. (line 203-204)

5 Please clarify whether the carbon increases were just due to the

addition of the biochar or to the addition of biochar and fixation. I refer the authors to the study by Weng et al Nature Climate Change; 7, (5) 371-376

Thanks for your review and helpful suggestion. We have made clarify for the carbon increases were not only just due to addition of the biochar but also biochar fixation. (line 336-343)

6 So if total C increased in the soil why did CO₂ increase. Where is the extra carbon coming from?

Thanks for your review. We believe the carbon comes from the biochar. Biochar input can directly affect the formation and content of soil organic matter. This promotion mainly attributes to the contribution of biochar itself containing stable and unstable components of the organic matter. Stable components of biochar can stabilize soil carbon pools when imported into soil and have long-term storage effects¹⁻³, and unstable component of biochar such as aliphatic compounds can directly increase the soil organic carbon storage capacity in the solid form^{4, 5}; aliphatic compounds gradually lose their peripheral structure and are released into the soil, directly decomposed by soil microbes, and then converted to MBC and DOM, thereby increasing the organic carbon pool of soil.

7 The discussion does not really include the most recent reports on how biochar builds up stable organo mineral layers. I suggest the authors read Archanjo et al (2017) Nanoscale analyses of the surface structure and composition of biochars extracted from field trials or after co-composting using advanced analytical electron microscopy. *Geoderma* . 295,70-79 . and Weng et al

Thanks for your review and helpful suggestion. These papers are very helpful to us. We have added biochar builds up stable organo mineral layers to our discussion part, and add these reference into our manuscript. (line 336-343)

8 Who else has found that *Lactococcus* plays a major role in soil carbon pool and what other bacteria have other researchers found that increase carbon when biochar is added?

Thanks for your review. Except for Hu we mentioned in our manuscript, Tang found *Lactococcus* have a close relationship with soil respiration⁷.

⁸. Somers research shows soil fertility of organic agricultural soils can be related to the presence of *Lactococcus*⁹. And Hu found *Lactococcus* potential impacts on carbon sequestration on wetlands⁶. And we have already added to our manuscript (line 400-405). There are also some bacteria were found increased when biochar added, including *Pseudomonas* spp. , *Lysobacter* spp. ¹⁰and *Bacillus* spp. ¹¹.

Reviewer #2:

No comments.

Reference:

1. B. S. Archanjo, M. E. Mendoza, M. Albu, D. R. G. Mitchell, N. Hagemann, C. Mayrhofer, T. L. A. Mai, Z. Weng, A. Kappler, S. Behrens, P. Munroe, C. A. Achete, S. Donne, J. R. Araujo, L. Van Zwieten, J. Horvat, A. Enders and S. Joseph, *Geoderma*, 2017, **294**, 70-79.
2. Z. Weng, L. Van Zwieten, B. P. Singh, E. Tavakkoli, S. Joseph, L. M. Macdonald, T. J. Rose, M. T. Rose, S. W. L. Kimber, S. Morris, D. Cozzolino, J. R. Araujo, B. S. Archanjo and A. Cowie, *Nat Clim Change*, 2017, **7**, 371-+.
3. N. Hagemann, S. Joseph, H. P. Schmidt, C. I. Kammann, J. Harter, T. Borch, R. B. Young, K. Varga, S. Taherymoosavi, K. W. Elliott, A. McKenna, M. Albu, C. Mayrhofer, M. Obst, P. Conte, A. Dieguez-Alonso, S. Orsetti, E. Subdiaga, S. Behrens and A. Kappler, *Nat Commun*, 2017, **8**.
4. C.-H. Cheng, J. Lehmann, J. E. Thies, S. D. Burton and M. H. Engelhard, *Org Geochem*, 2006, **37**, 1477-1488.
5. A. Zhang, R. Bian, G. Pan, L. Cui, Q. Hussain, L. Li, J. Zheng, J. Zheng, X. Zhang and X. Han, *Field Crops Research*, 2012, **127**, 153-160.
6. Y. Hu, L. Wang, Y. S. Tang, Y. L. Li, J. H. Chen, X. F. Xi, Y. N. Zhang, X. H. Fu, J. H. Wu and Y. Sun, *Soil Biol Biochem*, 2014, **70**, 221-228.
7. Y. S. Tang, L. Wang, J. W. Jia, X. H. Fu, Y. Q. Le, X. Z. Chen and Y. Sun, *Soil Biol Biochem*, 2011, **43**, 638-646.
8. Y. S. Tang, L. Wang, J. W. Jia, Y. L. Li, W. Q. Zhang, H. L. Wang and Y. Sun, *Ecol Eng*, 2011, **37**, 1638-1646.
9. E. Somers, A. Amke, A. Croonenborghs, L. van Overbeek and J. Vanderleyden, *Poster presentation, Montpellier, Plant Research International*, 2007.
10. Haitao, X.T. 2018. Improvement of Tobacco-planted cinnamon soil by biochar and its microecological mechanism. Ph.D. Thesis, Henan Agricultural University, Zhengzhou, China.
11. P. Yang, Q. Li, J. Jiao, W. Wang, X. Chen and Y. Duan, *Journal of Yunnan Agricultural University*, 2015, **30**, 50-57.

Appendix C

Journal: *Royal Society Open Science*

Manuscript ID: RSOS-181499.R2

Manuscript Title: Biochar application on paddy and purple soils in southern China: soil carbon and biotic activity

Author(s): Shen Yan, Zhengyang Niu, Aigai Zhang, Haitao Yan, He Zhang, Kuanxin He, Xianyi Xiao, Nianlei Wang, Chengwei Guan, Guoshun Liu

Dear editor and reviewers,

Thank you very much for giving us the opportunity to revise and resubmit our manuscript. We highly appreciated the professional and illuminating comments provided by you. We have tried our best to revise the manuscript accordingly. We provide detailed, point-by-point responses in the following pages. Note that the reviewer' comments are presented in black font, and our responses are in blue font. Please let me know if you have any further questions and suggestions. We hope the revision will be satisfactory.

Sincerely,

Shen Yan and Guoshun Liu

Reviewer: 1

Comment 1: Much improved paper. Further editing of the english would make the paper more readable.

Response: Thanks for your review. This manuscript has been edited by a native speaker.

CERTIFICATE OF ENGLISH EDITING

This document certifies that the paper listed below has been edited to ensure that the language is clear and free of errors. The logical presentation of ideas and the structure of the paper were also checked during the editing process. The edit was performed by professional editors at Editage, a division of Cactus Communications. The intent of the author's message was not altered in any way during the editing process. The quality of the edit has been guaranteed, with the assumption that our suggested changes have been accepted and have not been further altered without the knowledge of our editors.

TITLE OF THE PAPER
Biochar application on paddy and purple soils in southern China: soil carbon pools and biotic activity

AUTHORS
Shen Yan, Zhengyang Niu, Aigai Zhang, Haitao Yan, He Zhang, Kuanxin He, Xianyi Xiao, Nianlei Wang, Chengwei Guan, Guoshun Liu

JOB CODE
WAROY_2_5

Signature

Vikas Narang,
Senior Vice President,
Operations-Author Services, Editage

Date of Issue
May 23, 2019

Editage, a brand of Cactus Communications, offers professional English language editing and publication support services to authors engaged in over 500 areas of research. Through its community of experienced editors, which includes doctors, engineers, published scientists, and researchers with peer review experience, Editage has successfully helped authors get published in internationally reputed journals. Authors who work with Editage are guaranteed excellent language quality and timely delivery.

Contact Editage

Worldwide request@editage.com +1 877-334-8243 www.editage.com	Japan submissions@editage.com +81 03-6868-3348 www.editage.jp	Korea submit- korea@editage.com 1544-9241 www.editage.co.kr	China fabiao@editage.cn 400-005-6055 www.editage.cn	Brazil contato@editage.com 0800-892-20-97 www.editage.com.br
--	--	---	--	---

Comment 2: They refer to the non aromatic carbon as unstable. This is

not correct as some of the non aromatic water soluble organic compounds have similar structures to humic acids and polyphenols. I would recommend using the term labile

Response: Thank you for the comments. We have already changed the word "unstable" into "labile" .

Reviewer: 2

No comments.